



# The role of preconditioning for extreme storm surges in the western Baltic Sea

Elin Andrée[1,2,*], Jian Su[1], Morten Andreas Dahl Larsen[2], Martin Drews[2], Martin Stendel[1], and Kristine Skovgaard Madsen[1]

[1]Danish Meteorological Institute, Lyngbyvej 100, 2100 Copenhagen, Denmark
[2]Technical University of Denmark, Dept. of Technology, Management and Economics, Produktionstorvet, Building 424, 2800 Kgs. Lyngby, Denmark
[*]Present Address: SMHI Göteborg, Sven Källfelts gata 15, 426 71 Västra Frölunda, Göteborg, Sweden

**Correspondence:** Elin Andrée (elin.andree@smhi.se)

**Abstract.** When natural hazards interact in compound events, they may reinforce each other. This is a concern today and in the light of climate change. In the case of coastal flooding, sea-level variability due to tides, seasonal to inter-annual salinity and temperature variations or larger–scale wind conditions modify the development and ramifications of extreme sea levels. Here, we explore how prior conditions influence peak water levels for the devastating coastal flooding event in the western Baltic Sea in 1872. By imposing a range of antecedent conditions in numerical ocean model simulations, we quantify the change in peak water levels that arise due to alternative preconditioning of the sea level before the storm surge. Our results show that different preconditioning could have generated even more catastrophic impacts. As an example, a simulated increase of 36 cm compared to the 1872 event was seen in Køge just south of the Danish capital region – a region that was already severely impacted. The increased water levels caused by the alternative water mass distributions propagate until encountering shallow and narrow straits, thereafter the effect vastly decreases. Adding artificial increases in wind speeds to each study point location reveals a near-linear relationship with peak water levels for all Western Baltic locations highlighting the need for good assessments of future wind extremes. Our research indicates that a more hybrid approach to analysing compound events, and readjusting our present warning system to a more contextualised framework, might provide a firmer foundation for climate adaptation and disaster risk management.

## 1 Introduction

Several authors have recently demonstrated the importance of considering the compoundness of extreme events and suggested that such events may become more likely due to climate change (AghaKouchak et al., 2020; Santos et al., 2021; Vogel et al., 2021; Zscheischler et al., 2018). They include a range of natural hazards like floods and storms, whose impacts may be enhanced or lessened by antecedent conditions that either interact directly with the event or affect the vulnerability of exposed areas (Bischiniotis et al., 2018; Bradstock et al., 2009; Johnson et al., 2016; McMillan et al., 2018; Raymond et al., 2020). The time scales of such "preconditioning" can vary from days to months or even years. For example, the exceptional 2018 European wildfire season that severely impacted Northern Europe was locally preceded by above-average temperatures and



abnormally dry (e.g. vegetation) conditions in most places, some extending back several months and some all the way back to 2017 (Commission et al., 2019). It was also generally exacerbated by favourable wind conditions and high temperatures during the summer. As compared to the average of 2008-2017, some countries like Norway, Sweden, Finland, Germany and the Czech Republic, therefore suffered a doubling or more of the number of recorded fires in 2018 (Commission et al., 2019). Similar examples involving different time scales include landslides that are predated by extensive soil erosion caused by e.g. rainfall or snowmelt (Hilker et al., 2009), as well as overland flooding induced by heavy rain that is exacerbated by falling on top of a very wet period, e.g. with saturated soils and filled water reservoirs (Hendry et al., 2019).

Management of the current and future risks of natural hazards often relies on a combination of learning and extrapolating from past extreme events, modelling and climate change projections (Dangendorf et al., 2021; Frederikse et al., 2020; Harjanne et al., 2017; Travis and Bates, 2014). However, while the history of meteorological observations is long, modern-era instrumental measurements only date back to the founding of the first meteorological institutes in the later part of the 19th century. As a result, comprehensive observations of low-probability high-impact events are generally scarce and limited to recent decades (Calafat and Marcos, 2020; Hallin et al., 2021; Jacobsen et al., 2021). In contrast, longer records include only the observed maxima, e.g. maximum observed water levels, inundation depths, precipitation intensities or wind speeds. Correspondingly, extremes inferred from model simulations are mainly compared to observations in their ability to reconstruct said maximum values and not their contexts (Marcos et al., 2015).

Storm surges and extreme sea levels are one of the main threats to people and property along coastlines (Brown et al., 2018; Buchanan et al., 2017; Hallegatte et al., 2013; Vousdoukas et al., 2020; Wahl et al., 2017). Generally, high water levels are associated with low-pressure weather systems, resulting in strong winds pushing seawater towards the shore and water levels exceeding the range of the astronomical tides. However, the wind effect is only one of several factors influencing high water levels' development, maximum elevation, and duration. Other essential factors include sea-level variations due to tides (Arns et al., 2020), seasonal or inter-annual salinity and temperature variations, large-scale pressure fluctuations, dynamic water interactions with basin geometry and bathymetry (especially for marginal seas), and the initial distribution of seawater within a basin (Pugh, 1987). In combination, these factors can lead to both heightened and lowered surge levels.

Coastal flood risk assessments are generally based on local extreme sea level statistics derived from time series of tide gauge measurements with lengths varying from a few decades to more than 100 years. Based thereof, extreme sea levels and their associated recurrence periods may be predicted using different variants of extreme value analysis (Coles et al., 2001; Thorarinsdottir et al., 2017; Wahl et al., 2017). Similarly, future extreme sea level statistics may be obtained by analysing modelled sea levels within a future time slice, e.g. 2071-2100, and contemporary scenario assumptions (Masson-Delmotte et al., 2021; Oppenheimer et al., 2019). It has been proposed that hydrodynamic models may be needed to refine flood risk assessments at regional to local scales. For example, Vousdoukas et al. (2016) suggest that by accounting for water level attenuation due to land surface roughness, the estimated flood exposure decreases (inundation extent and depth) and hence also the estimated damages (Vafeidis et al., 2019). Likewise, several authors have recently addressed the potentially disproportional risks from compound coastal flooding, e.g., caused by a combination of storm surge and heavy rainfall (Bevacqua et al., 2019) or a surge combined with high river discharge (Couasnon et al., 2020), and the challenges for risk management concerning





compound events in our study area (Modrakowski et al., 2022). Conversely, the role of preconditioning for the development of extreme sea levels has so far received less attention (Weisse and Weidemann, 2017). Here, we exemplify the potential

influence of preconditioning of the Baltic Sea for an extreme sea-level event in the western Baltic. The Baltic Sea is a marginal sea of the Atlantic Ocean characterised by complex coastlines. Its connection to the North Atlantic, via the North Sea and the shallow and narrow Danish Straits, suppresses much of the sea-level variability coming from the North Atlantic. Instead, this flow restriction introduces other types of sea-level variability that may exacerbate extreme sea levels induced by storms. Atmospheric forcing can redistribute water between the different sub-basins in the Baltic or change the overall volume and

thereby the filling level on time scales of weeks (Samuelsson and Stigebrandt, 1996; Weisse and Weidemann, 2017). Likewise, oscillations related to the semi-enclosed nature of the Baltic Sea known as seiches (Leppäranta and Myrberg, 2009; Pugh, 1987) are found to contribute to sea-level variability. However, these are not yet fully understood (Weisse et al., 2021). The characteristic time scales for these oscillations have been estimated to roughly a day based on basin-wide (Wubber and Krauss, 1979), and sub-basin wide (Jönsson et al., 2008) premises.

The importance of considering the contribution from filling level and seiches to Baltic sea-level anomalies has previously been highlighted by Weisse and Weidemann (2017), who analysed sea level data from a high-resolution tide-surge model driven by an atmospheric reanalysis. In their 64-year hindcast, high filling level (FL-H, defined as periods where the sea level near Landsort, Sweden, remain at least $15\,\mathrm{cm}$ above the local long-term mean for a minimum of twenty days (Mudersbach and Jensen, 2010)) occurred on average sixty days per year. During these conditions, relatively lower wind speeds were needed to

generate high sea levels. Weisse and Weidemann (2017) also showed that seiche contributions to peak water levels exceeded $10\,\mathrm{cm}$ in one-third of cases at the station Wismar on the German Baltic Sea coast.

In this study, we revisit the disastrous 1872 (western) Baltic Sea storm surge (Clemmensen et al., 2014; Colding, 1881; Rosenhagen and Bork, 2009), which stands as the worst storm surge on record experienced in the western Baltic Sea (Hallin et al., 2021). During this event, an unparalleled wind forcing from the northeasterly–easterly sector over a large expanse of

the Baltic Sea (Rosenhagen and Bork, 2009) generated exceptional water levels, up to $3.5\,\mathrm{m}$ above average, affecting areas in Denmark, Germany and Sweden with catastrophic impacts (Colding, 1881; Hallin et al., 2021; Jacobsen et al., 2021). At least 271 persons drowned, and about 15000 lost their homes (Kiecksee et al., 1972; Petersen and Rohde, 1977). 427 sailing ships (15 of them Danish) and 23 steam ships stranded or sank, mainly along the eastern shores of the Danish islands of Zealand and Falster (Bureau Veritas, 1872).

Interestingly, the Baltic Sea filling level in the weeks before 13 November 1872 was fairly moderate. The main objective of this paper is to answer the question: What extreme water levels would have been obtained as a consequence of the 1872 storm if the antecedent conditions were different? We explore this research question using a set of numerical ocean model simulations that all arrive at states driven by the atmospheric conditions of the 1872 storm surge event. The differences between the simulations arise as we change the point of departure (i.e. the antecedent conditions) of the simulations to assess the effect of

different kinds of preconditioning on the resulting water levels. The regional atmospheric conditions during the 1872 storm have previously been reconstructed by Rosenhagen and Bork (2009) at the German national meteorological service Deutscher Wetterdienst (DWD). Their product yields higher maximum wind speeds that better agree with local observations than what is





generated in lower resolution global reanalysis (Feuchter et al., 2013). Here, the regional reconstruction is used as forcing for our simulations.

In the following, we specifically compare the model simulations of 1872 with three alternative scenarios with more favourable preconditioning to quantify a range of implications of an "1872-like" storm. The substitute antecedent conditions are based on realistic simulations of contemporary sea-level events. In addition, we carry out a second set of simulations where we amplify the wind speeds used as input to the ocean model. The purpose of these simulations was to assess the combined effect of storm and preconditioning enhancement on peak water levels. In Sect. 2, we outline the atmospheric conditions of the 1872
storm surge, the experimental design, data sources, and the ocean model setup. Section 3 presents our results, and Sect. 4 the discussion and conclusions.

## 2   Methods and data

The following section describes the atmospheric conditions during the original experiment, i.e. the unperturbed simulation of the 1872 storm surge as reconstructed by our model system. We denote this experiment O. Section 2.2 describes our three
variant preconditioning scenarios (which we denote FL1, FL2 and S) and the physical conditions behind these cases.

### 2.1   Case study: The 1872 event

On 13 November 1872, catastrophic flooding took place along the southwestern Baltic Sea coasts. Water levels exceeded previous records by far, and no flood event has come even close to the 1872 event since then. Water levels reached $3.38\,\mathrm{m}$ in Lübeck, $3.40\,\mathrm{m}$ in Travemünde and Eckernförde, $3.30\,\mathrm{m}$ in Kiel, $3.49\,\mathrm{m}$ in Schleswig and $3.27\,\mathrm{m}$ in Flensburg (Petersen and
Rohde, 1977). For Danish coastlines, Jacobsen et al. (2021) provide trend free sea-level estimates based on the comprehensive collation of contemporaneous oceanic and atmospheric information by Colding (1881). Relative to the mean sea level in the year 2020, the water level reached $2.90\,\mathrm{m}$ at Køge and increased westward to more than $3.5\,\mathrm{m}$ by the Danish mainland (Jacobsen et al., 2021).

Favourable conditions for a storm flood are generated when westerlies transport large amounts of water through the Danish
Straits and into the Baltic Sea. A dangerous rise in the water level at the Baltic Sea coasts of Germany and Denmark can occur if the wind subsequently changes to a northeasterly direction. This mechanism was already discussed by Baensch (1875). Therefore, it is necessary to consider the atmospheric situation two weeks before the event when reconstructing the 1872 storm surge and similar events.

### 2.1.1   Atmospheric conditions

Between 1 and 11 November, low pressure was found over Scandinavia and the Norwegian Sea. Strong winds from westerly to southwesterly directions caused a net transport of water through the Danish Straits and into the Baltic Sea. The maximum cumulative transport at Cape Arkona on the island of Rügen (54.7°N 13.4°E) peaked on 9 November (Rosenhagen and Bork, 2009). On 10 November, the weather pattern changed dramatically. A low crossed central Europe on a quite unusual track





from northwest to southeast, while pressure rose sharply over Scandinavia. Consequently, the winds shifted from southwest
to northeast, and the piled-up waters in the eastern Baltic Sea could flow to the southwest. This situation – low pressure
over central Europe, high pressure over Scandinavia and a maximum pressure gradient over the southwestern Baltic Sea –
prevailed during the next three days, with both the high and the low intensifying further. In the morning of 13 November, the
high over central Scandinavia had an unusually high sea level pressure of $1047\,\mathrm{hPa}$, whereas the low with a core pressure of
$990\,\mathrm{hPa}$ was located over the border region of Saxony, Prussia and Bohemia. As a consequence, the northeasterly storm over
the southwestern Baltic reached full gale force. With the weakening of both pressure centres, the strong winds died down, and
water levels fell.

### 2.1.2 Data sources

The atmospheric conditions driving the development that culminated in the 1872 storm surge can be retrieved from a global re-
analysis based on synoptic pressure observations (Compo et al., 2011). However, more local data is available than is included in
global reanalyses. For our control simulation of the 1872 storm surge (denoted O), we, therefore, utilised two different sources
of atmospheric forcing. First, to spin up the ocean model, we used forcing from the 20th Century Reanalysis in its most recent
version 20CRv3 (Slivinski et al., 2019) for a simulation spanning the years 1871 to 1873. The 20CRv3 data set is available in
three-hourly resolution and $75\,\mathrm{km}$ grids (Slivinski et al., 2019) (https://psl.noaa.gov/data/gridded/data.20thC_ReanV3.html).
Second, we used a regional, gridded reconstruction with higher spatial ($0.5°$ grids) and temporal (hourly) resolution (Rosen-
hagen and Bork, 2009) in the days preceding and during the storm surge event. This data set was supplied by the German
national meteorological service (DWD). It is based on a more extensive set of observations and captures the very intense wind
conditions during the event more accurately than the coarse, global reanalysis (Feuchter et al., 2013). For the analysis of the
1872 event, we have access to a substantial amount of local and regional data, notably from Germany. Observations have also
been preserved from other nations, many of which had already established weather services. In Denmark, Niels Hoffmeyer,
the first director of the newly founded Danish Meteorological Institute, reconstructed sea level pressure fields from numerous
observations that the DMI had collected.

As pointed out in the previous section, one of the preconditions for the catastrophic flooding was the period of strong
westerlies prior to the event that transported large amounts of water into the Baltic Sea. Therefore, the period from 1 to 14
November 1872 was considered in the reconstruction by Rosenhagen and Bork (2009), and the investigated area covered the
northeast Atlantic and northern Europe as far east as the Baltic states. We used this data set when available, i.e., from 06:00 1
November until the storm surge abated almost two weeks later.

The methods for generating the detailed 1872 atmospheric reconstruction is described in Rosenhagen and Bork (2009). Here
we give a brief overview of the concept behind the manipulation. Generally, we are interested in observations of sea level
pressure and wind direction and speed. From there, we can reconstruct the two-dimensional (geostrophic) wind fields that are
required to run our ocean model. In practice, geostrophic wind fields can be determined by triangulation and compared to
the wind observations. This construction is achieved by assuming an equilibrium between the Coriolis force and the pressure
gradient force (Alexandersson et al., 1998). An extrapolation needs to be done to obtain winds at $10\,\mathrm{m}$ height since the pressure

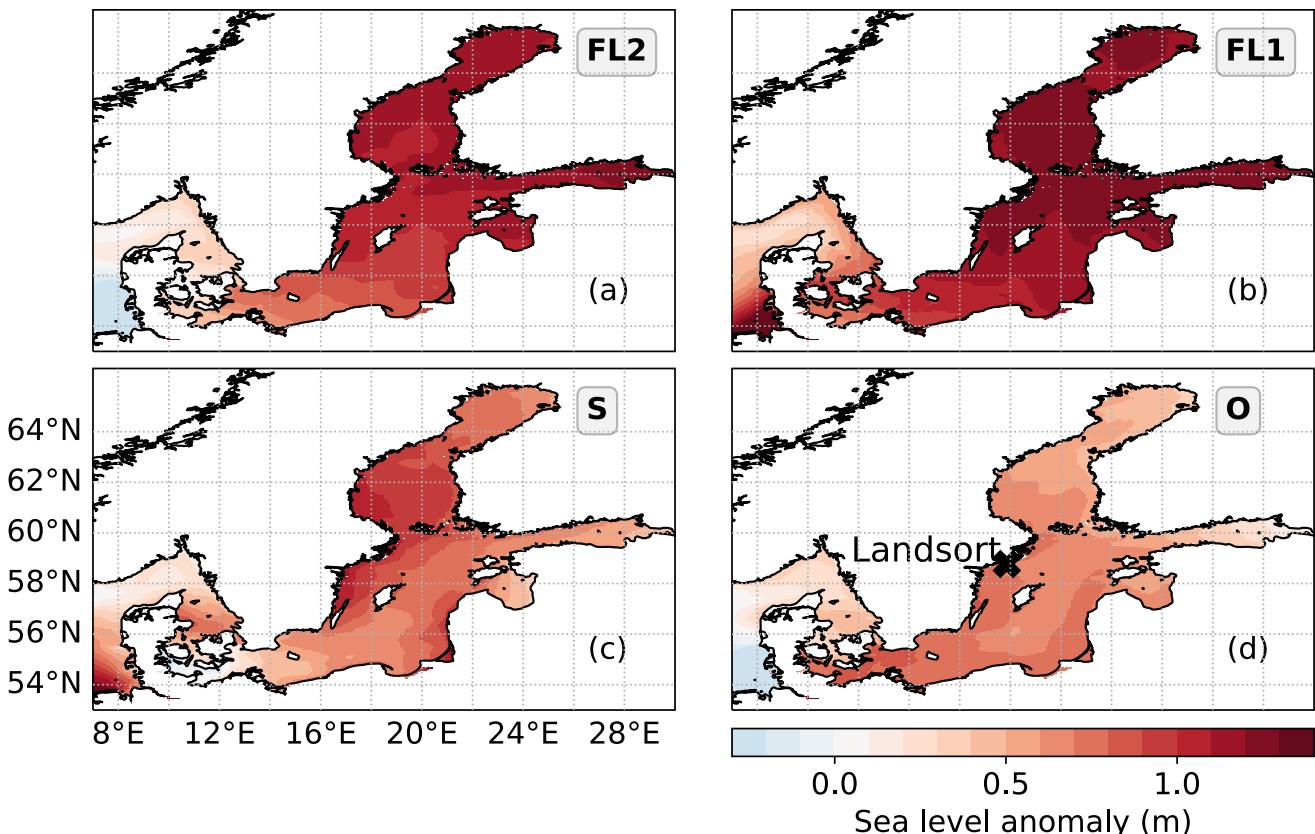

**Figure 1.** The sea level anomaly field that corresponds to the maximum water level at Landsort for each simulation (see Fig. 2 for time series). Panel (d) shows the unperturbed case (O) from 14:00 11 November 1872. Preconditioning for the sea level anomalies in panels (a) and (b) were obtained from an ocean hindcast (Andrée et al., 2021). The time is adjusted to match case O. Therefore, the time steps shown correspond to 9 November at 19:00 (FL2) and midnight (FL1), respectively. Case S (panel (c) uses the same conditions as O, except that the atmospheric forcing between midnight on 9 November and up until 15:00 on 12 November 1872 are replaced by the corresponding times from 1st to 4th January 2017. The panel represents 04:00 on 12 November. Panel (d) shows the location of station Landsort, which is used to estimate the filling level.

fields have been reduced to sea level. Such extrapolations can be accomplished using empirical formulae. Many approaches have been suggested for this purpose, but common to them all is that they are quite dependent on the thermal layering of the

160 lower troposphere, which we do not know. Further, this approach does not directly take into account frictional effects. Both factors can be approximated by using the distance from the sea, dependent on the wind direction.



## 2.2 Alternative preconditioning

To investigate scenarios of how altered antecedent conditions could have affected the development of the 1872 storm surge, we conducted three different experiments with alternative preconditioning. Two of the cases (FL1 and FL2) represent instances of high filling levels within the majority of the Baltic Sea. Case S incorporates a seiche effect. The data and methods for generating the scenarios are described in Sect. 2.2.1. The selection and physical conditions surrounding the instances are described in Sect. 2.2.3–2.2.4.

### 2.2.1 Scenario construction

As previously mentioned, the filling level of the Baltic Sea in November 1872 was fairly moderate. To demonstrate the implications for extreme sea levels if the Baltic had been preconditioned differently, we formed scenarios by imposing the atmospheric forcing of 1872 onto three alternative cases where the water mass distributions were different (Fig. 1). The water level at Landsort (58.8°N, 17.9°E) (location marked in panel (d), Fig. 1) is commonly used to represent the filling level in the Baltic Sea because it is close to the nodal line of the Baltic (Feistel et al., 2008; Lisitzin, 1974; Matthäus and Franck, 1992; Weisse and Weidemann, 2017). The development of the Landsort water level for the respective simulations are shown in Fig. 2. In addition to showing the Landsort water level, Fig. 2 indicates the periods we use as preconditioning (i.e. alternative antecedent conditions) for the perturbed cases and the Landsort water levels corresponding to the snapshots in Fig. 1.

Scenarios FL1 and FL2 utilise monthly archived initial conditions from a regional ocean hindcast (Andrée et al., 2021). We forced the ocean model with the same regional reanalysis as the ocean hindcast (i.e. the Uncertainties in Ensembles of Regional Re-Analyses (UERRA) HARMONIE/V1 data set (Ridal et al., 2017)) from the initialisation at the beginning of the respective month until the desired preconditioning state was reached (see Sect. 2.2.3–2.2.4). Horizontal bars in Fig. 2 mark these periods. The atmospheric forcing was thereafter switched directly to that of the high-resolution, 1872 reconstruction corresponding to 9 November. From then on and throughout the rest of the simulations, the atmospheric forcing that drives cases FL1 and FL2 is identical to the unperturbed (O) case. Differences in the dynamic development for each scenario are therefore solely due to perturbations of the initial state. The periods that utilise unperturbed atmospheric forcing from the 1872 event are indicated by solid colours (horizontal bars, Fig. 2). Case S is identical to O until midnight 9 November 1872, when the forcing was switched to that of midnight 1 January 2017. This forcing was utilised up until 15:00 on 4 January when it was switched to the corresponding time from 12 November 1872. In effect, we replaced approximately 3.5 days of case O to incorporate a seiche effect in S.

### 2.2.2 Case FL2 - 13 March 1990

As a complement to using Landsort's water level we did a spatial integration of sea-level anomalies eastward of 13°E to assess the Baltic filling level. The highest value corresponds to the Landsort maximum on 30 January 1983 and is described in Sect. 2.2.3 (case FL1). The second highest event constitutes our case FL2, initialised on 13 March 1990.



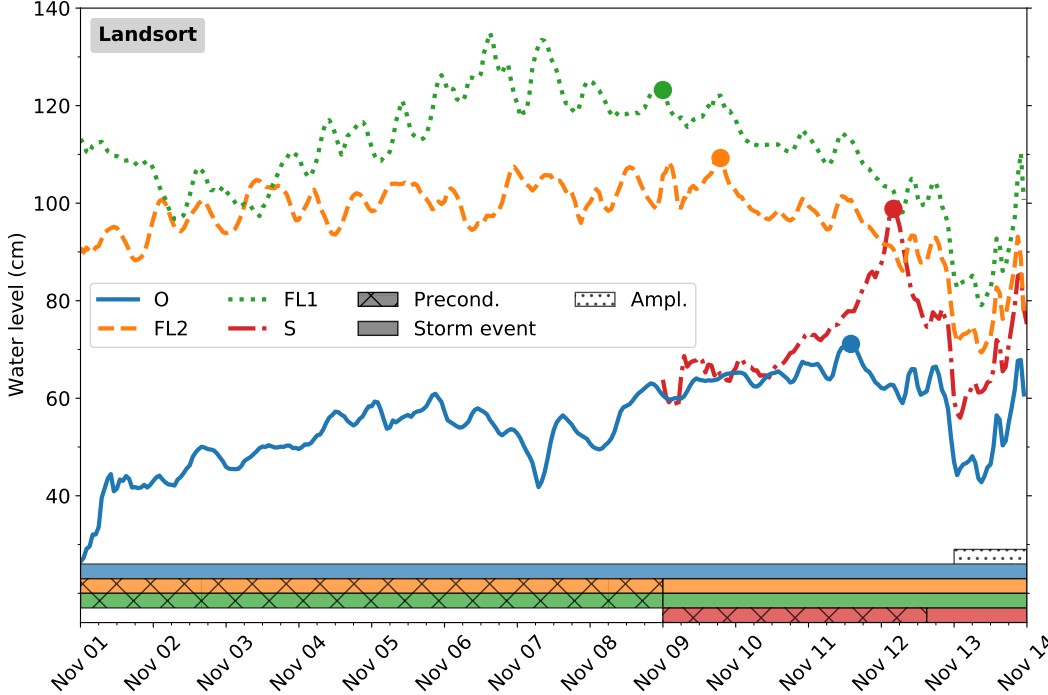

**Figure 2.** Preconditioning of the original (O) and alternative scenarios (FL2, FL1 and S) illustrated by the water level at station Landsort. The dots show the Landsort water levels corresponding to the sea level distributions in Fig. 1. Horizontal bars indicate the respective preconditioning periods (Sect. 2.2.1).

The year 1990 started unusually warm and was dominated by winds from southerly to westerly directions. Intense activity from low-pressure systems over the North Atlantic resulted in a succession of storms and frontal passages tracking over the North Sea. The strong zonal winds with intermittent episodes of northwesterly winds pushed water masses into the Baltic Sea. From 21 February until 13 March, when case FL2 was initialised, the water level at Landsort steadily increased. Sea level elevations were high overall but lower in the Bothnian Bay and Baltic Proper than in case FL1.

### 2.2.3 Case FL1 - 1 February 1983

Case FL1 occurs in the aftermath of the highest observed water level at Landsort (Wolski et al., 2014). The atmospheric conditions leading up to this event constituted an extensive period of mild and wet weather with strong, zonal winds. The water level at Landsort started rising within the first few days of December. On 18 January, a low-pressure system that generated northwesterly, hurricane-strength winds along the Danish North Sea coastlines tracked from the north of the UK and eastward towards the central Baltic Sea. During its passage, the relative water level at Landsort reached its highest observed value in an almost 136-year long record. In the last week of January, southwesterly to westerly winds over the North Sea and the south to central Baltic Sea were mainly between 10 to $20\,\mathrm{m\,s^{-1}}$. On 31 January, the Baltic Sea experienced winds of only a few metres





per second, as a new low-pressure system was moving in over the northern UK. The wind-driven volume increase in the Baltic Sea generated persistent, elevated sea levels throughout most of the Baltic Sea (Fig. 1). The FL1 simulation was initialised from the state of the ocean at midnight on 1 February. At that time, the water level at Landsort had lowered slightly but remained exceptionally high (Fig. 2).

### 2.2.4 Case S - 4 January 2017

We constructed case S to incorporate the dynamics of a so-called silent surge event that impacted the western Baltic Sea in 2017 (She and Nielsen, 2019). The Danish Storm Council classified the silent surge as a 50-year event (i.e. 2% or less chance of occurring in a given year) along Danish coastlines, despite only moderate and far-field wind forcing that was mainly distributed over the central Baltic Sea (She and Nielsen, 2019). A key component in this development was the preconditioning, with an elevated water level in the Baltic Sea and the Kattegat, in comparison to the southwestern Baltic Sea (She and Nielsen, 2019). This much more temporary and dynamic preconditioning is blended into case S.

Case S utilised the same atmospheric forcing and initial conditions as O, except for the period between midnight 9 November and 15:00 on 12 November, which was replaced by midnight 1 January to 15:00 on 4 January. This period was used to alter the preconditioning compared to O. Leading up to midnight 9 November, southerly to southwesterly winds had pushed water masses northward into the northern Baltic Sea, generating a substantial sea-level gradient between the northern and southwestern ends. The onset of 1 January 2017 forcing started with northerly winds of around $10\,\mathrm{m\,s^{-1}}$ over the North Sea, southwesterly winds over the Baltic Proper and weaker, northerly winds over the northern Baltic basins. Water masses that had been piled up in the Bothnian Bay had started to move southwards. The wind turned northwest with around $10\,\mathrm{m\,s^{-1}}$ wind speeds over the Baltic and slightly higher over the North Sea. The wind field over the North Sea intensified and turned more westerly as a low-pressure system reached Norway. It tracked over the central Baltic Sea, following a southeasterly trajectory while generating northwesterly winds of around $20\,\mathrm{m\,s^{-1}}$ on its backside. At the forefront of the system, the southwesterly to easterly winds pushed water masses north and westward. North of this low-pressure system, a high-pressure system intensified. This weather pattern generated northeasterly winds of about $20\,\mathrm{m\,s^{-1}}$ over the Baltic Sea, along with northerly winds over Kattegat.

The atmospheric forcing that generated the 2017 surge continues to unfold for several hours after we switch back to the 1872 forcing (Sect. 2.2.1). In this way, the scenario captures the piling-up in the central Baltic Sea that sets the stage for the 2017 surge. It also captures the atmosphere's development into a persistent pressure distribution similar to 12 November 1872. From then on, we utilise the more intense and longer-lasting winds of 1872. In the observed development, relatively weaker, northeasterly winds over the Baltic Sea persisted for some hours more, thereby adding to the severity of the 2017 surge.

### 2.3 Wind forcing amplification

In addition to the experiments detailed above, we conducted simulations of cases FL1, FL2 and O to amplify the wind forcing. These experiments aimed at illustrating whether changes in the wind forcing would generate feedback by either dampening or enhancing the influence of preconditioning in the perturbed scenarios relative to the control. We achieved this intensification





of the wind forcing by increasing the wind speed by 20% (FL1, FL2 and O) or 30% (FL1 and FL2 only) in the atmospheric
forcing corresponding to 13 November 1872. This period is indicated by a dotted, horizontal bar in Fig. 2.

### 2.4    Storm surge modelling

For the storm surge simulations, we used the regional, 3D, baroclinic ocean circulation model HIROMB-BOOS Model (HBM)
for the North Sea and Baltic Sea (Berg and Poulsen, 2012; Kleine, 1994; She et al., 2007). For a detailed description, see
e.g. (Berg and Poulsen, 2012; Poulsen and Berg, 2012). HBM employs a two-way nesting scheme, allowing for the exchange
of mass and momentum between the coarse and finer grids to resolve the complex flow structures of water exchange in the
transition zone between the brackish Baltic Sea and the more saline North Sea. The coarse grid domain has a spatial resolution
of $5.5\,\mathrm{km}$ and 50 vertical layers. The fine-grid domains are located in the German Bight and the inner Danish waters (transition
zone between the North and Baltic Sea). They have $1.9$ and $0.9\,\mathrm{km}$ spatial resolution with $24$ and $52$ vertical layers, respectively.
We used climatological river run-off data obtained from the Hydrological Predictions for the Environment model for Europe
(E-HYPE) (Donnelly et al., 2016). HBM has been used for a wide range of applications in, e.g., climate and hindcast studies
(Andrée et al., 2021; Fu et al., 2012; Madsen, 2009; Su et al., 2021; Tian et al., 2016), for assessing wind-driven sea-level
sensitivity (Andrée et al., 2022) as well as for local marine management efforts of coastal estuaries (Murawski et al., 2021) and
radioactive tracer studies (Lin et al., 2022). The present version was used for operational storm surge forecasting at the Danish
Meteorological Institute between 2013 and 2018.

### 255    3    Results

As already stated, we use case O as a reference simulation for the 1872 storm surge. The peak water levels obtained for this
simulation agree with historical records within a few decimeters along the Danish coastlines but are overestimated by almost
a meter at Travemünde. Overall, the results from case O confirm that the simulation is an appropriate point of departure for
exploring alternative developments of the 1872 storm surge event.
Figure 1 shows the initial distribution of water masses in the Baltic Sea corresponding to the 1872 storm and the three
alternative scenarios. As illustrated, cases FL2 and FL1 are characterised by overall increased volumes in the Baltic Sea. In
contrast, case S is mainly characterised by a temporary piling-up of water in the Gulf of Bothnia. For both FL2 and FL1,
the filling level is consistently higher than during the 1872 storm surge (O). Conversely, S is roughly similar in magnitude to
O but exhibits a somewhat different mass distribution. Figure 2 shows the corresponding water levels measured at Landsort,
which is often used to indicate the general Baltic Sea filling level (Feistel et al., 2008; Matthäus and Franck, 1992; Weisse and
Weidemann, 2017). The Landsort water level reflects the volume of water that could potentially flow back and cause floods in
the western Baltic Sea and the inner Danish seas upon release. The timestamps on Fig. 2 are adjusted so that the development
of cases FL2, FL1 and S matches that of the unperturbed event. As also shown in Fig. 1, cases FL2 and FL1 start with very high
water levels at Landsort (Fig. 2) in comparison to the unperturbed event. At the end of the preconditioning period, the difference
between these cases amounts to about $15\,\mathrm{cm}$. This shift remains after the onset of the 1872 forcing (9 November), and they



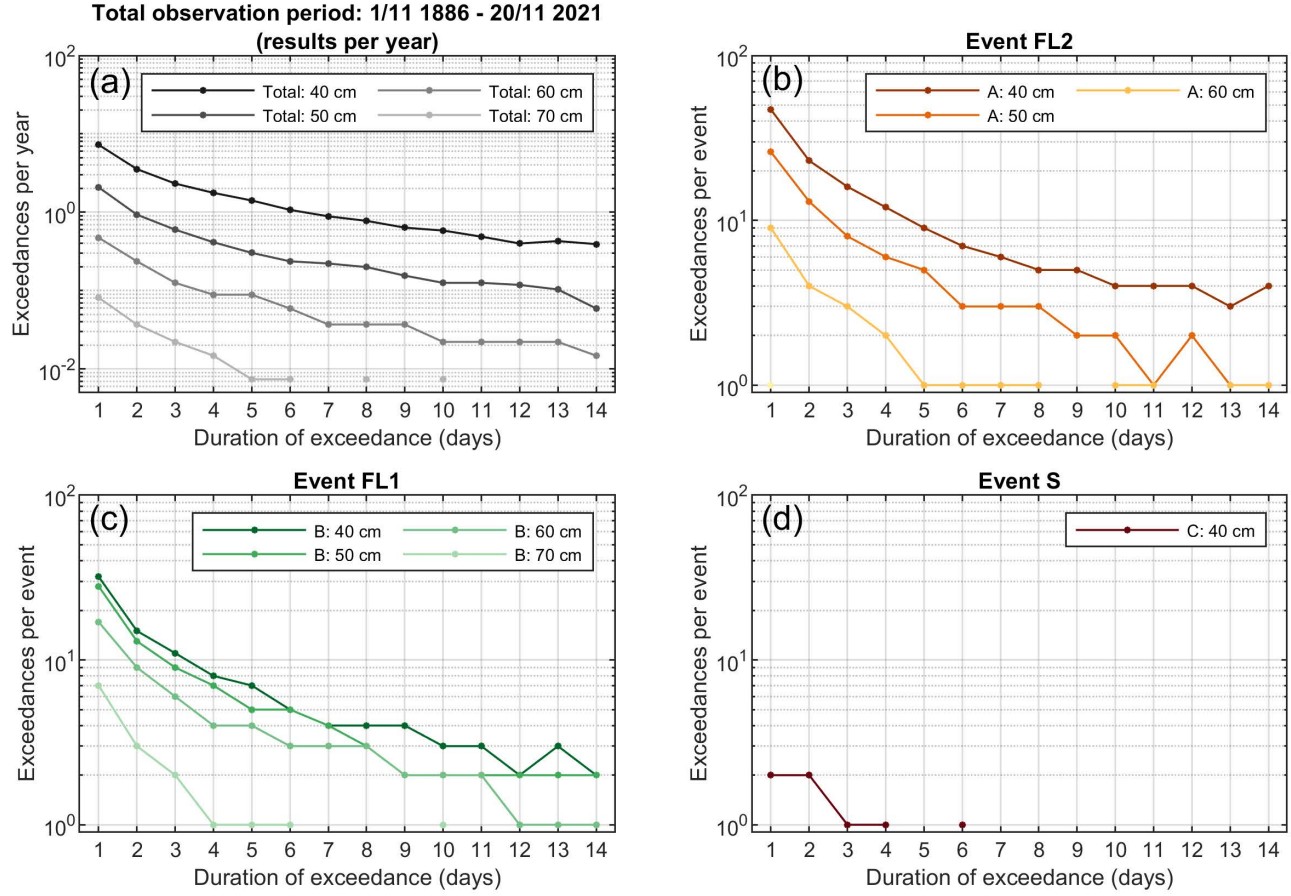

**Figure 3.** Frequency of specific durations (1-14 days) for water levels of 50-80 cm. Plot (a) is for the entire period with observation data (1886-2021 - results provided per year), and plots (b)-(d) are for the specific events FL2, FL1, and S, respectively (results per event). Plots (b)-(d) have a similar y-axis range. Data are from the Swedish Meteorological and Hydrological Institute (SMHI) Open data service (SMHI, 2021).

display very similar temporal patterns onward. This similarity can also be seen in case O regarding sub-daily oscillations. Cases FL2 and FL1 continued to be the highest throughout the event among the four cases presented here. The seiche event (case S) is identical to case O until the modification of the initial conditions on 9 November. Rather than the slow processes that bring about the high filling levels in cases FL2 and FL1 (Sect. 2.2.3–2.2.2), the preconditions for case S develop rapidly in

just a little over a day. Even though the forcing only differs from case O for a few days, the water level reaches 27 cm higher at Landsort due to the characteristics of this preconditioning. The 1872 event, case O, maintains a Landsort water level of around 60 cm until the sharp decrease, shared by all events, during the night between 12–13 November. At the time of this drop in


water level, the atmospheric forcing is identical for all cases, which result in nearly identical water level reductions of 21 to 22 cm across all four cases.

Due to the connection between high water-level events in the western Baltic Sea and the associated filling level of the Baltic Sea in general (Weisse and Weidemann, 2017), we assess the entire observation period (1886-2021) and each scenario for the relationship between sea levels and the corresponding duration. For this analysis, the Landort site is used since the water level here is a good proxy of the general filling level of the Baltic Sea. Further adding to this suitability, Landsort observations are available as far back as 1886 from the Swedish Meteorological and Hydrological Institute (SMHI, 2021).

Specifically, we here calculate the frequency in which mean water levels, in 10 cm steps and aggregated to different durations (1-14 days), occur. This aggregation is done per year for the entire time series and per event for each of the events, defined by the general curve breakpoint for the onset and ending of each event (108, 138 and 83 days for events FL2, FL1 and S, respectively). From Fig. 3 it is, for example, seen that an average three-day sea level of at least 60 cm occurs three times during FL2, six times during FL1 and does not occur for event S (panels b-d). The same water-level threshold and duration occur on

average 0.13 times per year (panel a).

From the cumulative distribution function (not shown), we find that 99.0% of the observations occur in the −50 to 50 cm interval and that 1, 10 and 100 year return periods correspond to hourly water levels of approximately 75.7 cm, 85.5 cm and 93.5 cm. Based on Fig. 3 and these return period statistics, the magnitude of water levels corresponding to FL1 and FL2 reflect relatively rare and extreme events, whereas event S is a high but not rare event. On this note, however, the one-year return

period level at Landsort accounts for 81% of the 100-year return period level, keeping in mind the close relation to the general Baltic Sea filling level. Therefore, relatively high filling levels are seen at frequent intervals.

The freshwater content in the Baltic Sea means that there is a northward tilt of the sea level throughout the Baltic Sea. This characteristic results in a discrepancy between modelled values and observed relative water levels at Landsort, which is why we here choose not to reflect scenario preconditioning levels in terms of return period rates.

Figure 4 shows the effect of the different preconditioning on the resulting maximum water levels in the western Baltic Sea. We subtracted the maximum values from the unperturbed case (O) from the maximums for cases FL2, FL1 and S to highlight spatial differences. The sea-level tilt between the northern- and easternmost basin ends versus the southern Baltic was most pronounced in the unperturbed representation. The maximum water level at Landsort occurred as these water masses gradually were flowing south and westwards, reducing the water level in the north and east and causing it to rise throughout

the southwestern Baltic (panel (d), Fig. 1). The alternative preconditioning results in altered peak water levels throughout the southwestern Baltic Sea, as seen in Fig. 4. Of these, case FL2 results in the highest water levels by far. The peak water levels reach values in the general order of 0.3-0.45 m above the 1872 (case O) reference, with the largest differences seen as a piling-up south of the Swedish coastline where the water masses encounter shallow depths. In a very narrow bay parallel to the German northeast coastline, the difference exceeds 0.5 m. In descending order, case FL2 is followed by case FL1 and case S

showing corresponding residuals, relative to case O. FL2 results in values of 0.2 to 0.3 m and display similar spatial patterns. For case S, on the other hand, differences of 0.25 to 0.3 m are mainly confined to the northeastern German coast, eastward of the narrow passageway between Germany and Denmark (Fehmarn Belt). One interesting feature of this case is that the signal




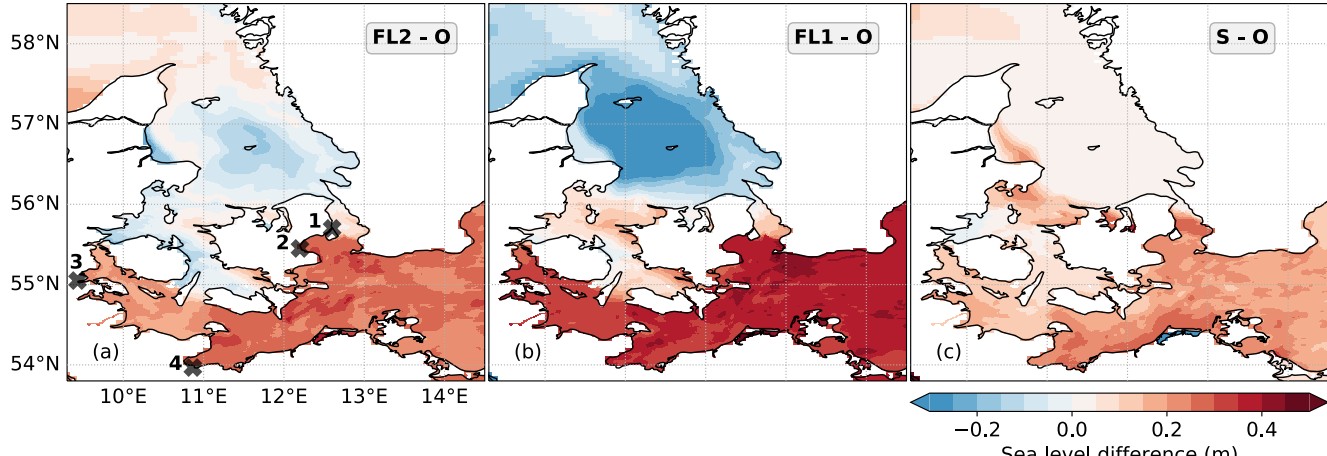

**Figure 4.** The effect of alternative preconditioning on the 1872 storm surge. The panels show the difference between the maximum sea level obtained with alternative preconditioning and the maximum sea level obtained with the unperturbed preconditioning (O). Panel (a) shows the locations of København (Copenhagen, the Danish capitol) (1), Køge (2), Aabenraa (3) and Travemünde (4).

**Table 1.** Summary of the simulated peak water levels for the different experiments (Sect. 2). The unperturbed simulation (O) numbers are given in absolute values. For the remaining scenarios, the values shown indicate the difference to the unperturbed simulation (O's values subtracted). The Landsort column represents the maximum water level after 9 November (marked with dots in Fig. 2) and is included here for comparison. The experiments FL2, FL1 and S, utilise the same atmospheric forcing as O but has different preconditioning. The scenarios denoted + 20% are the same as the respective O, FL2 and FL1, except that the wind speed was increased by 20% on 13 November.

| | Preconditioning | | Peak water level (cm) | | |
|---|---|---|---|---|---|
| Name | Landsort (cm) | København | Køge | Travemünde | Aabenraa |
| O | 71 | 114 | 252 | 425 | 385 |
| FL2 | + 38 | + 2 | + 28 | + 27 | + 20 |
| FL1 | + 52 | + 10 | + 36 | + 35 | + 32 |
| S | + 27 | + 26 | + 20 | + 21 | + 13 |
| O + 20% | Same as O | + 47 | + 108 | + 151 | + 158 |
| FL2 + 20% | Same as FL2 | + 49 | + 142 | + 181 | + 171 |
| FL1 + 20% | Same as FL1 | + 60 | + 153 | + 188 | + 183 |

of sea-level elevation extends into the Sound, past the very shallow threshold (Darss Sill, minimum depth of $8\,\mathrm{m}$) separating Denmark's biggest island from Sweden. Up to $0.3\,\mathrm{m}$ higher water levels occur in the region of the Danish capital and Sweden's third-biggest city. For all cases, the three straits of Øresund, Storebælt and Lillebælt enforce drastically reduced residual levels, and the corresponding levels in Kattegat even show a negative amplitude for cases FL2 and FL1, with residual levels down to approximately $-0.3\,\mathrm{m}$ for the former of these.






**Table 2.** Time of the peak water levels reached (see Table 1). Absolute timestamps are retrieved from the unperturbed simulation (O). For the remaining scenarios, the values shown indicate the difference in minutes compared to the unperturbed simulation (O's timestamps subtracted).

| | Peak time (min) | | | |
|---|---|---|---|---|
| Name | København | Køge | Travemünde | Aabenraa |
| O | Nov 13, 07:10 | Nov 13, 08:10 | Nov 13, 08:50 | Nov 13, 13:30 |
| FL2 | -10 | 10 | -10 | 0 |
| FL1 | -30 | 10 | -10 | -10 |
| S | -40 | 10 | 10 | 0 |
| O + 20% | 10 | 20 | 30 | 40 |
| FL2 + 20% | 0 | 30 | 20 | 30 |
| FL1 + 20 % | 0 | 50 | 20 | 30 |

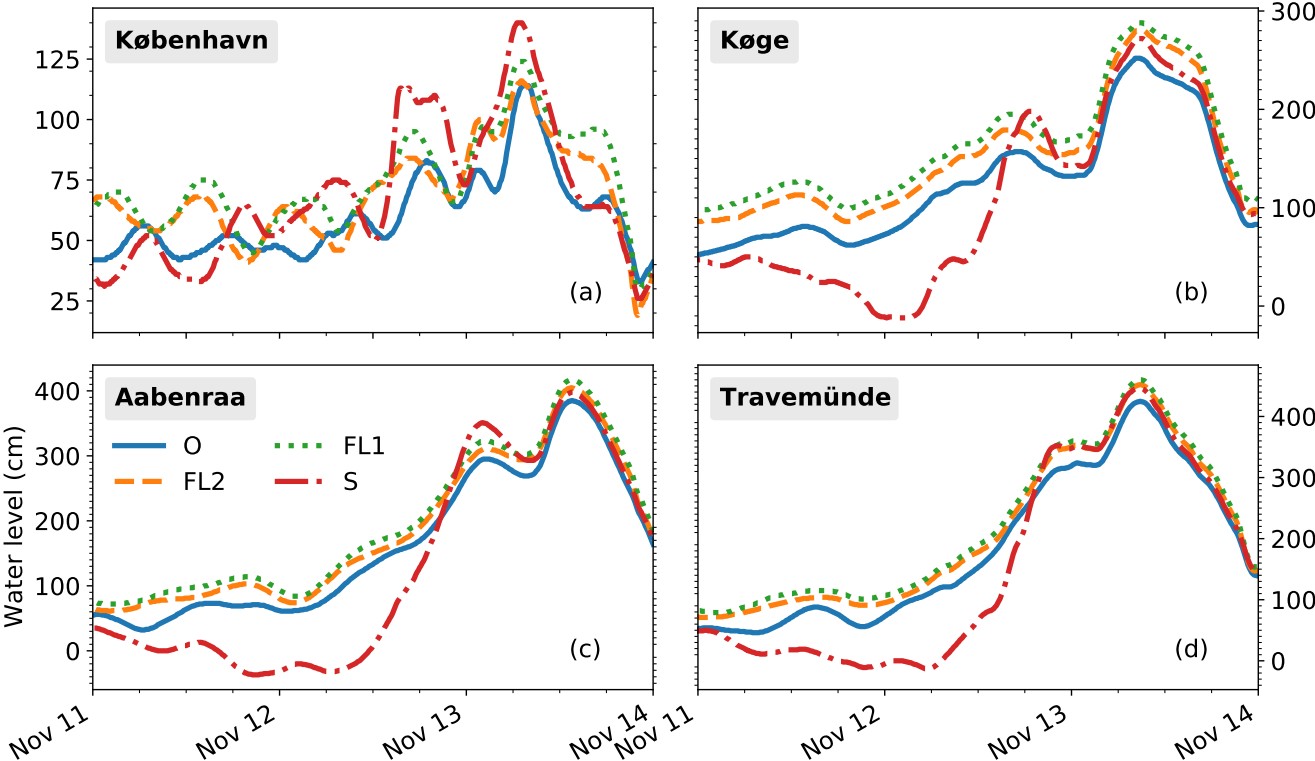

**Figure 5.** The effect over time of alternative preconditioning for the 1872 storm surge. The panels show how the water levels develop over time for the unperturbed case (O) and the three alternative preconditioning scenarios (FL2, FL1 and S) at four different locations. Notice the differences in the y-axis scale.





The maximum water levels (Table 1) and temporal water level developments (Fig. 5) are shown for four different stations distributed around the western Baltic Sea (locations marked in panel (a), Fig. 4). Referring to the O case, the timing of maximum

levels occur within 1 h 40 min for København (Copenhagen), Køge and Travemünde (Table 2). In contrast, Aabenraa, located along the Jutland east coast in the westernmost part of the Baltic Sea, has a peak 6 h 20 min after København, which has the earliest of the other three peaks. The alternative preconditioning result in higher peak water levels with differences ranging between 2 to 36 cm for all locations (Table 1). For comparison, the water level at Landsort was between 27 and 52 cm higher than O across the other scenarios. Between Køge and Copenhagen, the maximum water levels differ dramatically given the

30 km distance between them, with peak levels of 2.52 to 2.88 m for Køge and 1.14 to 1.40 m for Copenhagen (Table 1). Case FL2 exhibits the highest value for Køge, whereas case S is the highest for Copenhagen. In addition, Køge has a longer peak duration than Copenhagen. The fact that the Copenhagen time series is measured from the northern part of the city highly influences these results, as this location is located north of the shallow sill at the southern entrance of the Sound. Therefore, these results mainly reflect inner-Copenhagen sea levels, whereas the suburbs of Copenhagen facing towards the south are

likely to experience sea levels more comparable to those for Køge. Peak water levels for Aabenraa and Travemünde vary between 4.25 to 4.60 m and 3.85 to 4.17 m respectively, with the same order of cases as for Køge, whereas the peak duration to a higher degree resembles that of Copenhagen.

To investigate the combined effect of stronger winds and enhanced preconditioning for the 1872 event, we amplified the wind fields used to force the ocean simulations. The amplification was restricted to 13 November, and we used a fixed factor over the

entire wind field. The results from intensifying the wind speed by 20% (cases FL2, FL1 and O) and 30% (cases FL2 and FL1) are shown in Fig. 6. Amplification of the wind speed resulted in increased peak water levels with an almost linear response (Fig. 6), which seems to indicate that at least for the peak values, any dynamic changes to the water-mass distribution induced by the enhanced wind are marginal. At Copenhagen, 20% wind speed amplification resulted in 40 to 41% (up to 0.5 m) higher water levels, and 63 to 65% (up to almost 0.8 m) for a 30% increase in the wind speed. Køge had a slightly higher response

for 20% wind speed increase (41 to 43%, up to 1.14 m) and lower for 30% (61 to 63%, 1.76 m). Corresponding increases for Aabenraa reached 36 to 41% (up to 1.58 m) and 56 to 58% (2.33 m) and for Travemünde 33 to 36% (up to 1.54 m) and 52 to 53% (2.38 m), respectively.

As shown in Table 2, the higher wind speeds delay the peak water levels in all cases and for all locations, while the preconditioning itself shifts the peak times both backwards and forwards in time.

**4 Discussion**

In this paper, we quantify extreme water levels that could have been obtained as a consequence of the 1872 storm if the preconditioning was different. For this aim, we compared realistic model simulations of the 1872 storm surge with three alternative scenarios having more favourable preconditioning, drawn from reconstructions of contemporary sea-level events.


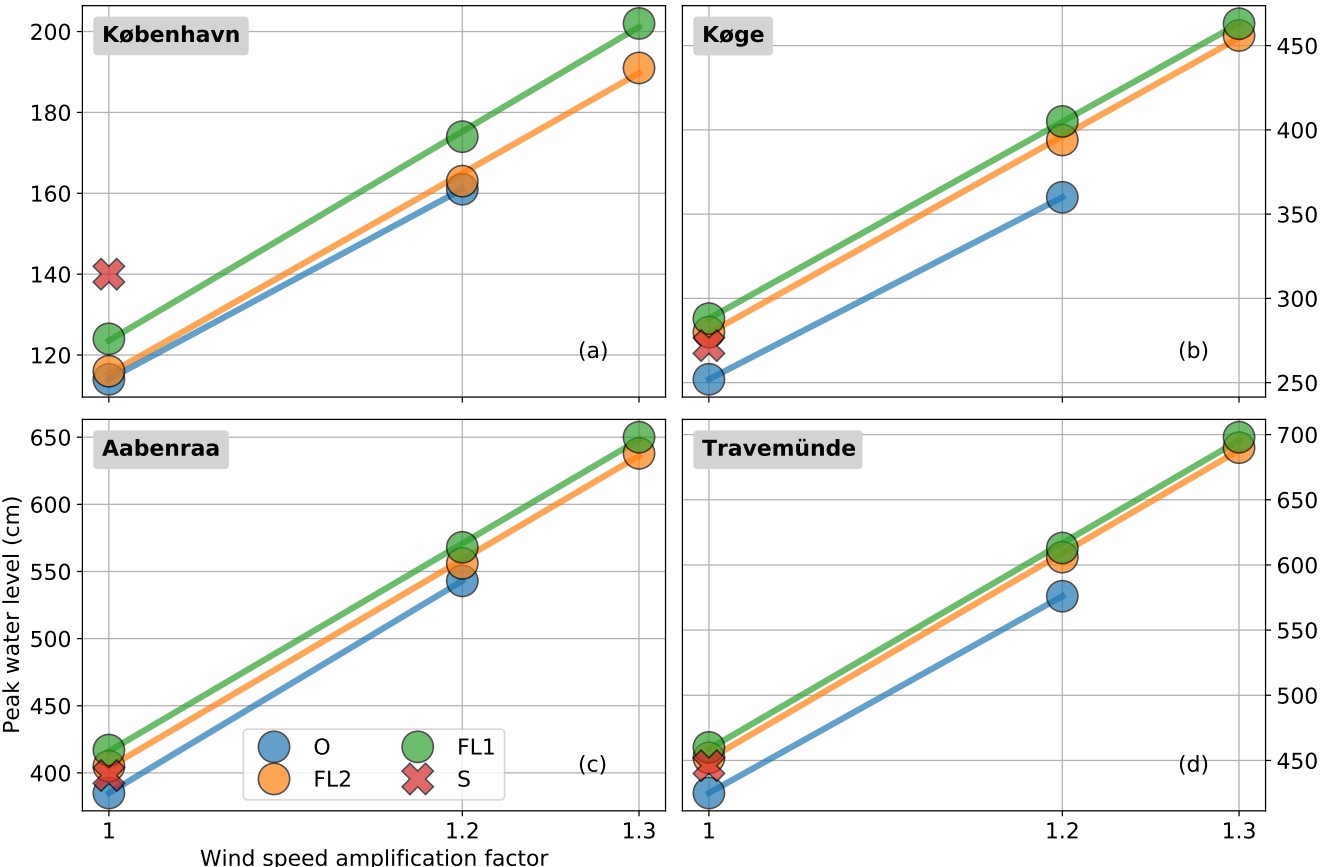

**Figure 6.** The effect of alternative preconditioning and wind speed intensification on peak water levels at four locations. The experiments O, FL2, FL1 and S with amplification factor 1 are the same as in Fig. 5. In addition, experiments O, FL2 and FL1 were run with wind speeds multiplied by a factor of 1.2 or 1.3 for FL2 and FL1 only. The lines show linear fits to the peak water levels for FL2, FL1 and S, respectively (filled circles). Note the different scales on the y-axes. See Sect. 2 for details on the respective intensification.

## 4.1 Effect of preconditioning

As shown in Table 1 and Fig. 4–5, the simulated extreme water levels for all three alternative scenarios overshoot the unperturbed values. When comparing S against FL2 and FL1, it is evident that the antecedent sea-level patterns (i.e. water mass distributions) also play a key role. The latter is also clearly seen from Fig. 5 regarding the local dynamics observed at København, Køge, Travemünde and Aabenraa. Depending on the exposed site of interest, our findings further suggest that the role of the preconditioning is crucial and that the effect is site-specific.

While this study intends to generate physically plausible scenarios, the way we modify the preconditions of the 1872 simulation by chaining together different physical events is purely synthetic. One could argue whether the combinations are physically





conceivable since they effectively represent unobserved events. All three of the cases FL2, FL1 and S could, however, be relevant in a climate change context.

Firstly, experiments FL2 and FL1 comprise high filling levels in the Baltic Sea. Figure 5 shows that the developments of these
events are highly similar. Due to the higher filling level (14 cm higher Landsort water level for FL1 than FL2), experiment FL1 results in higher peak water levels (Table 1). However, the difference is lower than the difference at Landsort. This discrepancy implies that the differences in peak water levels due to an increased volume in the Baltic Sea are not simple linear superpositions of the historic peak water levels and the volume difference as reflected by Landsort's filling level. An increased volume in the Baltic Sea will result from anthropogenic sea-level rise. Simply adding these drivers' contributions might overestimate the
future peak water levels.

Secondly, experiment S demonstrates a scenario where an extra-tropical cyclone (ETC) precedes the 1872 event, similar to the 2017 storm surge event. Such successive events could become more common under the climate warming scenarios because of more frequent atmospheric blocking. Atmospheric blocking events are prevailing, meteorological disturbances, commonly anti-cyclonic weather patterns, that deflect the large-scale, westerly flow in the mid-latitudes (Barriopedro et al., 2006; Stendel
et al., 2021; Woollings et al., 2018). Such flow-diversions can cause weather patterns to be blocked over a region, and the phenomenon is linked to various hydro-meteorological extremes (Rutgersson et al., 2021; Stendel et al., 2021). It has been proposed that atmospheric blocking events will occur more frequently in the future with climate change, particularly in the Northern Hemisphere (Nabizadeh et al., 2019). However, the understanding is hampered by the fact that climate models tend to underestimate the frequency of events (Zappa et al., 2014), and by a lack of knowledge of the feedback processes that may
arise due to potential future changes in atmospheric dynamics (Stendel et al., 2021).

We have discussed different approaches to preconditioning and their effect on extreme water levels. By comparing the 1872 and 2017 floods, it is clear that wind speed is also an essential factor. So the question arises whether the 1872 storm with altered preconditioning would constitute a "worst-case event". Two other storm events with a synoptic situation comparable to the 1872 event occurred in the 20th century. On 30–31 December 1904, the second-highest water level (1.43 m) for the period
1889-2007 was observed in Fredericia. In Travemünde (2.22 m) and Flensburg (2.33 m), high water levels were observed as well. Nine years later, on 30–31 December 1913, the highest recorded water level was recorded in Gedser. In Svendborg, water was 5-6 feet, and in Flensburg 2 m above normal. These events resemble the 1872 catastrophe with strong westerlies followed by storms from the northeast. From these events, global reanalysis-based estimates of the pressure gradients in the region are larger than in 1872. In both cases (1904 and 1913), this situation persisted for only a couple of hours. In addition, the wind was
from a slightly different direction, so not much damage was caused. However, a combination of the location and track of the 1872 low with pressure gradients of, e.g., the 1904 low over a more extended period, appears synoptically entirely possible. This would result in winds approximately 30% stronger than in the 1872 case.

It is not clear whether such a situation would happen more frequently under climate change conditions (Stendel et al., 2021). As Scandinavian highs often occur in autumn and winter, strong lows moving eastward over northern Germany could initiate
similar flooding events. With increasing temperatures, the atmosphere can bear more water vapour, so it appears possible that such a low could undergo vigorous development.



More speculatively, intense low-pressure systems originating from tropical cyclones have been observed over Great Britain. While this appears to have happened before (for example, the "Great storm of 1703"), such events could happen more frequently in a warmer climate. There is, however, no indication in model simulations that such kinds of events could occur more frequently than in the past.

## 4.2 Implications for risk management

The 1872 storm surge was exceptional in both intensity and loss of lives and is by far the worst event documented in the western Baltic Sea by strong historical evidence (Hallin et al., 2021; Jacobsen et al., 2021). In this respect, the event has frequently been used as the benchmark "worst-case scenario" for coastal floods in the Baltic. However, given the results discussed above, one could argue for using even more extreme values from a physical perspective. While undoubtedly the severity of the 1872 storm was driven by high wind speeds (above $30\,\mathrm{m\,s^{-1}}$), we show here that the filling level of the Baltic Sea can add several decimeters more. Given that large parts of the coastal areas in the western Baltic are low-lying this is a significant contribution. What remains is to quantify the present and future probability of such compound events. The 1872 storm surge has already been classified as a "low-probability, high impact event", so these would be even more rare events. Speculatively, extrapolating from Fig. 5 would have resulted in approximately the same flood levels as in 1872 by "swapping" 5% on the wind speed for optimal preconditioning, which perhaps would be *more* probable than the 1872 event itself.

Compared to 1872, the geography of the Baltic Sea region has significantly changed, and the number of people, assets and societal interests located along the coasts have increased as a result of general population growth and coastal urbanisation. Most of the major coastal cities along the Baltic Sea, including the low-lying capital region of Denmark that sits within the bottleneck passageway to the North Sea, have expanded in size and now critically rely on infrastructure that requires protection from seawater. Hence, the need for robust evidence on the risks of current and future storm surges has never been higher.

As mentioned above, extreme sea level statistics based on tide gauge measurements or future projections of extreme sea levels currently generally comprise the "standard" for engineers and risk managers to cope with the accumulating climate risks due to storm surges and sea-level rise. Our research shows that a more hybrid approach, combining extreme sea level statistics with state-of-the-art climate and ocean modelling, might be needed to understand the context of these extremes better. In this way, we can better account for the uncertainties and ensure a more robust platform for decision-making on climate change adaptation and disaster risk management. Such a hybrid approach could take the form displayed in this paper, where historical, well-described high water level events like the 1872 storm are revisited, and detailed numerical models are used to expand the uncertainty (e.g. by supplementing actual tide gauge measurements with perturbed model members) and to add to our physical understanding of how a combination of different factors lead to specific water levels.

## 4.3 Compound events under climate change and pre-warning system

As discussed previously, compound events, a combination of weather and climate extremes, are becoming more and more of a concern for many locations as the climate warms (AghaKouchak et al., 2020; Zscheischler et al., 2018). Those investigations, however, have not shed light on today's non-extreme events. The Intergovernmental Panel on Climate Change (IPCC) has



identified one of the primary climate change-related compound events as the consecutive occurrence of extreme or non-extreme events (Field et al., 2012). Climate change is altering storm surge events in our research area, and a non-extreme sea-level event today can have enormous consequences when it is paired with a subsequent, more severe storm surge event. As demonstrated by our results, a strong storm surge event in the western Baltic Sea area might have highly diverse effects depending on the initial filling conditions. However, our earlier attention was primarily drawn to the extreme cases, leaving the more common

events largely under-researched (Weisse and Weidemann, 2017). Preconditioning and storm surge duration were found to be critical in this research. Thus, the current early warning system is challenged.

The local storm surge early warnings are a vital tool for reducing the impact of events on human activities and preventing economic loss in the face of global warming scenarios. The current storm surge warning system is based on a straightforward peak-over-threshold method, with the threshold increasing in tandem with the rise in mean sea level. The issue with the existing

warning system is that it is difficult to contemplate storm surge events lasting an extended period of time. As a result, non-extreme events are typically overlooked while developing an early warning system. We demonstrated that an early warning system should consider far more time than the conventional forecast method now in use (5 days). As a result, it can account for preconditioning of an extreme storm surge event. Our findings provide guidance for future developments of early warning systems. Indeed, it is easier to provide warnings for the longer-duration volume build-up in the Baltic Sea than for the shorter

piling-up duration in experiment S. Early warnings for FL experiment situations that are well-designed allow for more time for planning and execution of hazard prevention and preparation measures.

## 5   Conclusions

Natural hazards and extreme events are contingent on the conditions before the event itself. However, historical records from before modern-era instrumental measurements often comprise only maximum values. Even when high-resolution observational

or model products are available, it has long been the practice to assess the peak values without considering their context through the application of extreme value analysis. Perturbations of one or several of the constituents that together comprise a natural hazard allow for explorations of alternative scenarios to take the context of the hazard into account. This study focused on perturbations of the preconditioning of an exceptional storm surge event in the mouth of a semi-enclosed, inland water body. The hazard is a high impact, low probability storm surge event that occurred in the western Baltic Sea in 1872. We

generated alternative developments of the extreme sea level hazard for this event by substituting the initial conditions. Here, we showed that alternative conditions could have further worsened the impacts by adding several decimetres to peak water levels. We suggest that a more hybrid approach of assessing the combined drivers and their contexts could provide a more robust foundation for climate adaptation and disaster risk management.

Furthermore, we find that the pressure gradient of this notorious storm has been exceeded by similar pressure patterns

on at least two occasions during the 20th century, although these events have been shorter lasting. When adding artificial intensification of the wind speed, our simulations yield almost linear responses of further water levels increases throughout the western Baltic Sea, highlighting the need for good assessments of wind extremes.



We stress that understanding and awareness of preconditioning increases the actionable information before a natural hazard. Earlier warnings allow for more time for planning and executing hazard prevention and preparation efforts.

*Author contributions.* EA: drafting of the manuscript and acquisition, analysis and interpretation of data. JS: acquisition of data and analysis and interpretation of data, editing of the manuscript. MADL, MS and MD: analysis and interpretation of data, editing of the manuscript. KM, MD and MADL: project supervision. All authors contributed to the manuscript and approved the submitted version.

*Competing interests.* The authors declare that they have no conflict of interest.

*Acknowledgements.* The work reported in this paper is part of a greater shared effort between the Technical University of Denmark (DTU)
and the Danish Meteorological Institute (DMI) with the aim of investigating the processes that lead to extreme storm surges through meticulous numerical atmospheric-oceanic modelling. Part of the funding was provided by the Danish State through the Danish Climate Atlas. A portion of the work was carried out within the "Extreme events in the coastal zone – a multidisciplinary approach for better preparedness" project, hosted by Uppsala University and funded by Swedish Research Council Formas.



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
