# Peer review of "The role of preconditioning for extreme storm surges in the western Baltic Sea"

_Natural Hazards and Earth System Sciences, 2022_

## Referee Comment (RC1)

**General comment**

While I think that the work by Andrée et al., may be of interest for the scientific community, in the present form of the manuscript it is difficult to appreciate all the analysis carried out and to understand what motivated the research.

In particular, the Authors should clearly explain why it would make sense to test different boundary conditions for modelling the 1872 storm in the Baltic Sea, and why these can be deemed reasonable (e.g., the combination of winds/water levels etc. can actually occur?).

Besides, it is not clear how the results presented in the paper may help hazard prevention plans that are often based on weather forecasting systems accounting for several environmental forcing (therefore already including compound effects).

**Specific comments**

Line 4: influence -> influenced

Line 5: since you are speaking of a numerical model, I would replace "antecedent" with "boundary"

Line 8: as far as I understood, the 36 cm was not seen, it was rather simulated. Also, I would replace "Danish capital region" with "Copenhagen region (Denmark)". Please note that not everybody is familiar with Baltic Sea's geography, so all the references to cities and places should be supported by figures and/or more detailed descriptions (see also below).

Line 19: why "or"? if the storms impact is enhanced it will in turn affect coastal vulnerability

Lines 21-29: the whole paragraph seems to be useless (please either remove it or shorten it)

Line 35: In contrast -> By contrast

Line 37: said?

Line 39: property -> properties

Line 65: please specify what the filling level is

Line 68: to be roughly equal to a day

Line 82: you usually do not start a sentence with numbers

Lines 87-90: this paragraph is unclear. Please rephrase it

Line 103: during the original experiment, i.e., -> for

Line 108: "bay far" is somewhat unformal for a scientific publication

From line 108 onward: all the cities mentioned should be shown in a figure so that the reader can spot where the different locations are, otherwise, it is very annoying to look them up on e.g., google earth every time

Line 117: why 2 weeks have to be considered?

Line 145: "the first director […] Institute" is a useless detail. Please remove

Figure 1: please extend the colorbar so that it includes also the left-side panels. Also, please change the marker color of Landsort as it is not easy to see it (green perhaps?). The Figure should be included after it is first referred to, i.e., in Section 2.2.1

Figure 2: I found this figure very difficult to understand

Line 244: remove brackets

Page 10, section 3: Travemunde and the Gulf of Bothnia are mentioned in the text but not shown in any Figure (note that Figure 4 is introduced at a later time)

Line 282: you already specified that water level at Landsort is a sound proxy; also, note that Landsort is misspelled

Lines 285-286: what does it mean that "mean water level" occurs? Actually, the whole paragraph starting at line 280 is rather confusing and it should be better rephrased

Line 292: are these results based on an Empirical Cumulative Distribution Funcion? If so, please specify

Figure 4: please extend the colorbar through all the panels. Also, the four markers are a bit difficult to spot. Consider changing their colors.

Line 313: Darss Sill?

Figure 6: I do not think that three or even two points are enough to fit a robust linear model

Line 355: how do you detect a physically plausible scenario? See the general comment

Line 380 onward: again, the locations are not shown in figures

Line 393-395: I am very confused. Within two lines, first you say that the events could happen more frequently, next you say there is no indication that these events could occur more frequently?!

Line 421: are becoming more and more of a concern -> are increasingly becoming a concern

Line 437: would it make sense to use forecasts longer than 5 days? Would they be reliable? Please at least discuss this trade-off

Line 447: context of the hazard -> hazard context

---

## Author Comment (AC1)

Dear Reviewer,

We thank you for your willingness to review our manuscript and for your constructive and thorough comments on the manuscript. Please find our responses to your comments in blue below.

Best regards,

Elin Andrée, Jian Su, Morten Andreas Dahl Larsen, Martin Drews, Martin Stendel and Kristine Skovgaard Madsen

**General comment**

*While I think that the work by Andrée et al., may be of interest for the scientific community, in the present form of the manuscript it is difficult to appreciate all the analysis carried out and to understand what motivated the research.*

Thank you for this comment. Based on the comments received from yourself and the other reviewer, we have tried to clarify both the analysis carried out (see below) and what motivated us. In terms of our motivation, we particularly try to highlight better the urgent need for assessing the context of natural hazards. Natural hazards like storm surges are often the result of a cascade of events where the respective links either intensify or reduce the magnitude of the hazard. Such a physical context is not considered in classical risk analysis such as calculations of recurrence frequencies from observed annual maxima or other collections of extremes. Rather, such analyses often only focus on the statistics of extremes while making critical assumptions about the representativeness of the underlying data sets. To avoid under- or overestimation of the risks and more generally to improve confidence in the results of these kinds of studies for the benefit of adaptation planning, it is necessary to be able to "explain" the numbers and the associated uncertainty estimates. This is particularly true for storm surges, which inherently represent a compound event comprised by wind and wave dynamics, tides, sea level, local site-specific conditions, etc. One way to investigate this is through event-based studies such as ours that strive to explain and use the physical findings of past events to address future risks (elements of the abovementioned explanation has been included in the Introduction).

*In particular, the Authors should clearly explain why it would make sense to test different boundary conditions for modelling the 1872 storm in the Baltic Sea, and why these can be deemed reasonable (e.g., the combination of winds/water levels etc. can actually occur?).*

Thank you for this valuable comment. From historical records and modelling reconstructions of the storm surge event of 1872, it is evident that the Baltic Sea filling level in the weeks preceding 13 November 1872 was quite moderate. On this background – and given that the 1872 storm often serves as a reference for, e.g. climate change adaptation around the Baltic Sea – it is relevant to ask whether the 1872 storm was really the worst possible event that could have happened in the western Baltic? Since the sea filling levels of the Baltic Sea exhibits natural variability with exceedances of the 1872 event, it is reasonable to assume that the initial sea filling levels (serving as boundary conditions for the storm) could have been higher than they were (and lower as well for that matter). Rather than using a set of synthetic and potentially inconsistent boundary conditions to represent higher filling levels, we choose to use the boundary conditions derived from historic events. Based on historic events, we try to answer the question whether the 1872 storm was the worst possible event in the western Baltic. One event of particular interest is the storm surge on 31 December 1904. For this storm, the pressure gradient in the western Baltic was about 1/3 larger than

for the 1872 event. With easterly winds (rather than from the northeast as in 1872), water levels ran up to the "top five" at several locations in Denmark, but remained well below the 1872 values (Jacobsen et al., 2021). Another event, which we used as preconditioning, is the "silent storm surge" (i.e., a storm surge without storm) of 4 January 2017 with high water levels, also among the "top five" at several locations (Jacobsen et al., 2021). This was due to a high water level in the days before the event. By definition, the observed combinations of wind and water levels during these previous events represent realistic conditions, and while none of them are exact "scalings" of the 1872 event, we argue that our modifications are physically plausible as they are well within the local range of natural variability. This extends to the question of the physical realism of the met-forcing scenarios, with the exception of the transition within the model simulations (when we change from the preconditioning forcing data to those of the reconstructed 1872 event) – a transition that the model handles robustly. The clarifications above are now included in the Introduction (Line 85-).

*Besides, it is not clear how the results presented in the paper may help hazard prevention plans that are often based on weather forecasting systems accounting for several environmental forcing (therefore already including compound effects).*

The issue with existing weather forecasting systems is that it is difficult to contemplate storm surge events lasting an extended period of time. Therefore, non-extreme events are typically overlooked. This is exactly the point in section 4.3. To clarify, we discuss this in more detail in the Abstract and Conclusion.

*"The current storm surge warning system is based on a straightforward peak-over-threshold method, with the threshold increasing in tandem with the rise in mean sea level. The issue with the existing warning system is that it is difficult to contemplate storm surge events lasting over an extended period of time. As a result, non-extreme events are typically overlooked while developing an early warning system. We demonstrated that an early warning system may have to consider far more time than the conventional forecast method now in use (5 days). "*

**Specific comments**

*Line 4: influence -> influenced*

Based on a comment from Reviewer #2 we have changed the phrasing to "may influence".

*"Here, we explore how prior conditions may influence peak water levels for the devastating coastal flooding event in the western Baltic Sea in 1872."*

*Line 5: since you are speaking of a numerical model, I would replace "antecedent" with "boundary"*
Agreed! We have rephrased the sentence:

*"We quantify the change in peak water levels that arise due to alternative preconditioning of the sea level before the storm surge by imposing a range of historically observed circumstances as boundary conditions to numerical ocean model simulations."*

*Line 8: as far as I understood, the 36 cm was not seen, it was rather simulated. Also, I would replace "Danish capital region" with "Copenhagen region (Denmark)". Please note that not everybody is familiar with Baltic Sea's geography, so all the references to cities and places should be supported by figures and/or more detailed descriptions (see also below).*

Thank you for this suggestion. The administrative region, which encompasses Copenhagen and North Zealand, is formally named "the Capital Region of Denmark". However, since we here refer to a

subset of this region, we have revised the manuscript as proposed to read "Copenhagen (Denmark) and surrounding areas".

We added Figure 1 to show a map of the study area.

[Figure]

*Figure 1: Map of the study area: A zoom in to the Danish Straits (panel (a)) and the Baltic Sea (panel (b)). The panels mark locations mentioned in the text. Red dots indicate locations that the analysis focusses on.*

**Line 19: why "or"? if the storms impact is enhanced it will in turn affect coastal vulnerability**

This is a good point. To clarify, we have rephrased it:

*"They include a range of natural hazards like floods and storms, the impacts of which may be enhanced or lessened by compounding conditions that interact directly with the event, hence affecting the vulnerability of exposed areas."*

**Lines 21-29: the whole paragraph seems to be useless (please either remove it or shorten it)**

We respectfully disagree with the reviewer. We believe that this section highlights the relevance of prior conditions and their effect on the intensity of many types of natural hazards, as well as the time spans that may need to be considered to capture the context of specific events.

**Line 35: In contrast -> By contrast**

Agreed. Revised.

**Line 37: said?**

Thank you. We have rephrased the sentence to exclude "said":

*"Correspondingly, the extremes inferred from model simulations are mainly compared to observations in their ability to reconstruct maximum values and not their contexts."*

**Line 39: property -> properties**

Agreed. Revised.

**Line 65: please specify what the filling level is**

We have now shifted the description of how the filling level is used to infer volume changes in the Baltic Sea to this location in the text:

*"Atmospheric forcing can redistribute water between the different sub-basins in the Baltic, or change the overall volume through water transport between the North and Baltic Seas, which may cause the sea level to vary on time scales of weeks (Samuelsson and Stigebrandt, 1996; Weisse and Weidemann, 2017). Volume changes are commonly inferred from the water level at Landsort (58.8°N, 17.9°E) because of its location close to the nodal line of the Baltic Sea and is referred to as the Baltic's filling level (Feistel et al., 2008; Lisitzin, 1974; Matthäus and Franck, 1992; Weisse and Weidemann, 2017)."*

**Line 68: to be roughly equal to a day**

Agreed - revised.

**Line 82: you usually do not start a sentence with numbers**

Agreed - revised.

**Lines 87-90: this paragraph is unclear. Please rephrase it**

As suggested by the reviewer, we have rephrased this paragraph:

*"The differences between the simulations arise as we change the initial sea-level patterns (i.e. the prior conditions) of the simulations. This sensitivity test allows us to isolate the effects of preconditioning on extreme sea levels resulting from a specific storm."*

**Line 103: during the original experiment, i.e., -> for**

Agreed - revised.

**Line 108: "bay far" is somewhat unformal for a scientific publication**

Agreed - rephrased:

*"Water levels greatly surpassed previous records, …"*

**From line 108 onward: all the cities mentioned should be shown in a figure so that the reader can spot where the different locations are, otherwise, it is very annoying to look them up on e.g., google earth every time**

Agreed - we added them in new Figure 1.

**Line 117: why 2 weeks have to be considered?**

This is very valid point. We meant to say "at least two weeks", and this has now been amended. The reasoning for considering at least two weeks relates to the relevant time scale for the filling level of the Baltic Sea.

**Line 145: "the first director […] Institute" is a useless detail. Please remove**

Agreed - revised.

**Figure 1: please extend the colorbar so that it includes also the left-side panels. Also, please change the marker color of Landsort as it is not easy to see it (green perhaps?). The Figure should be included after it is first referred to, i.e., in Section 2.2.1**

Agreed - revised.

**Figure 2: I found this figure very difficult to understand**

To clarify, we added a more thorough description in the caption, which now reads:

*"Preconditioning of the original (O) and alternative scenarios (FL2, FL1 and S) illustrated by the water level at station Landsort. The dots show the Landsort water levels corresponding to the sea level distributions in Fig. 1. Horizontal bars indicate the respective preconditioning periods; hashes indicate periods where the forcing in the alternative scenarios differs from that in O. See Sect. 2.2.1 for a description of the scenarios. The dotted bar indicates the period used to amplify the wind speed (see Sect. 2.3). "*

**Line 244: remove brackets**

Thanks.

**Page 10, section 3: Travemunde and the Gulf of Bothnia are mentioned in the text but not shown in any Figure (note that Figure 4 is introduced at a later time)**

We have now indicated these locations in the new Figure 1.

**Line 282: you already specified that water level at Landsort is a sound proxy; also, note that Landsort is misspelled**

Agreed, we have removed the reasoning for using Landsort and corrected the spelling.

**Lines 285-286: what does it mean that "mean water level" occurs? Actually, the whole paragraph starting at line 280 is rather confusing and it should be better rephrased**

Thank you for pointing this out. By "mean water level" we meant averaged water level during the period of exceedance. This is further clarified in the paragraph/caption now.

*"Due to the connection between high water-level events in the western Baltic Sea and the associated filling level of the Baltic Sea in general (Weisse and Weidemann, 2017), we assess the entire observation period (1886-2021) and each scenario for the occurrence pattern between elevated sea levels and their corresponding duration. For this analysis, we use Landsort observations, available from the Swedish Meteorological and Hydrological Institute (SMHI, 2021) and dating back to 1886.*

*Specifically, we calculate the frequency with which certain elevated water levels occur (in 10 cm steps and aggregated to durations of 1-14 days). This analysis is performed on a yearly basis for the entire time series and per event for each of the events. As an example, a three-day sea level of +60 cm occurs three times during FL2 and six times during FL1 but does not occur for event S (panels (b)-(d), Fig. 3). On average, the same water-level threshold and duration occur on average 0.13 times per year in the Landsort observational record (panel (a), Fig. 3)."*

**Line 292: are these results based on an Empirical Cumulative Distribution Funcion? If so, please specify**

Yes. We added this clarification.

**Figure 4: please extend the colorbar through all the panels. Also, the four markers are a bit difficult to spot. Consider changing their colors.**

Agreed - revised.

**Line 313: Darss Sill?**

Thanks.

***Figure 6: I do not think that three or even two points are enough to fit a robust linear model***

We agree and have added a reference to another study by the author team, where the same model setup was used for idealized simulations. In this study, we discussed the wind stress parameterisation and the formulation of a drag coefficient in the model setup in more detail.

*"Amplification of the wind speed resulted in increased peak water levels with an almost linear response (Fig. 6). This finding depends strongly on the model's wind stress parameterisation and drag coefficient, which was discussed in Andrée et al. (2022) for idealized simulations with the same model. The linear response seems to indicate that, at least for the peak values, any dynamic changes to the sea-level patterns induced by the enhanced wind are marginal. "*

***Line 355: how do you detect a physically plausible scenario? See the general comment***

As mentioned above, we used combinations of wind and water levels based on observations of previous events in the region. This guarantees the physical realism of the preconditioning scenarios, and since they fall well within the range of natural variability, we assume that they are physically plausible although they of course represent unobserved events (see also the answer to the general comment above).

***Line 380 onward: again, the locations are not shown in figures***

Noted. We added them in the new Figure 1.

***Line 393-395: I am very confused. Within two lines, first you say that the events could happen more frequently, next you say there is no indication that these events could occur more frequently?!***

We have now rephrased the paragraph to read:

*"More speculatively, intense low-pressure systems originating from tropical cyclones have been observed over Great Britain. While this appears to have happened before (for example, the ``Great storm of 1703''), from a physical perspective, such events could be expected to happen more frequently in a warmer climate. There is, however, no indication in model simulations that such kinds of events will occur more frequently than in the past."*

***Line 421: are becoming more and more of a concern -> are increasingly becoming a concern***

Agreed- revised.

***Line 437: would it make sense to use forecasts longer than 5 days? Would they be reliable? Please at least discuss this trade-off***

Thank you for this suggestion. It is difficult to say whether forecasts longer than five days would be reliable and provide added value without conducting a sensitivity analysis, which is beyond the scope of the current study. However, we agree with the reviewer that this topic is worthwhile to mention, and therefore the revised manuscript includes a new paragraph referring to recent developments in ECMWF's medium-range forecast.

*"We demonstrated that an early warning system should in principle consider far more time than the conventional forecast method now in use (5 days), i.e. to better account for the potential preconditioning of an extreme storm surge event. ECMWF began operational application of medium range forecasts (6-15 days) in 1979 (Bengtsson, 1985). With more than 40 years of experience, the medium-range forecast is becoming increasingly accurate, and recent advances in identifying the*

*growing errors in the long-range forecast have contributed to enhance the operating system's predictability (Lillo & Parsons, 2017; Matsueda & Palmer, 2018). Our findings provide …"*

Bengtsson, Lennart (1985), Medium-Range Forecasting at the ECMWF. Advances in Geophysics 28, 3–54. doi:10.1016/s0065-2687(08)60184-3

Lillo, S.P. and Parsons, D.B. (2017), Investigating the dynamics of error growth in ECMWF medium-range forecast busts. Q.J.R. Meteorol. Soc., 143: 1211-1226. https://doi.org/10.1002/qj.2938

Matsueda, M, Palmer, TN. Estimates of flow-dependent predictability of wintertime Euro-Atlantic weather regimes in medium-range forecasts. Q J R Meteorol Soc. 2018; 144: 1012– 1027. https://doi.org/10.1002/qj.3265

**Line 447: context of the hazard -> hazard context**

Agreed - revised.

---

## Author Comment (AC2)

Dear Reviewer,

We thank you for your willingness to review our manuscript and for your thorough and helpful comments on the manuscript. Please find our responses to your comments in blue below.

Best regards,

Elin Andrée, Jian Su, Morten Andreas Dahl Larsen, Martin Drews, Martin Stendel and Kristine Skovgaard Madsen

*This study is a nice example of exploration of potential changes to the existing catastrophic events in future climates. It is based on the perception that extremely dangerous situations in the Baltic Sea are usually formed by a sequence of episodes that are dynamically connected in time rather than a combination of basically random reactions of the sea to various forcing components that are governed by some extreme value distribution. This is a reasonable way forward in the Baltic Sea conditions where the sea level "climate" of several sub-basins may contain statistically almost impossible outliers.*

*The analysis is sound and professional. All aspects of the modeling efforts have been explained in detail so that even an inexperienced in modeling reader can enjoy the line of thoughts and catch the main points. The use of English and technical aspects of the manuscript are fine. The outcome is carefully justified and the formulated conclusions fully supported.*

*Therefore, I recommend the manuscript for publication basically as it is.*

Thank you very much!

*However, there are some fairly minor items, adjustment of which may make the presentation even better. Only one issue definitely needs clarification for inexperienced readers: sea level elevations propagate in many occasions as (long) waves, so what moves is wave energy rather than water mass.*

We agree with the reviewer and have changed our wording accordingly in the manuscript to clarify (including as indicated in the comments below).  Hence we have for example replaced the phrasing "water mass distribution" with "sea-level pattern" and now use "piling up" and "travelling" or "propagating" exclusively instead of "pushing" or other references to motion that would suggest displacement of mass rather than energy.

*Abstract, line 4: it would be better to say "prior conditions may influence".*

Yes; revised accordingly.

*Line 6: consider saying "certain" instead of "different".*

Yes; revised accordingly.

*Line 7: consider saying "increase in the water level of 36 cm".*

Yes; revised accordingly.

*Line 9: it is strongly recommended to say "water mass distributions propagate as (long) waves" (I guess this meant).*

Agreed – revised.

*Page 2, lines 40–46: it might be useful to mention also wave-driven set-up that may in some occasions provide up to 1/3 of the total surge.*

We have added a sentence about the wave setup:

*"Wave-driven setup from waves breaking in the shallow surf zone may comprise 20 to 30% or more of the total surge during energetic wind conditions (Lavaud et al., 2020; Woodworth et al., 2019)."*

*Page 4, line 95: it might be more appropriate to speak about "more unfavourable" preconditioning here, on line 114 and on page 15.*

Agreed – revised.

*Line 117: from the presentation it seems that "at least two weeks" would be more exact.*

Agreed – revised.

*Line 121: winds probably caused "intense net transport".*

OK – revised.

*Lines 121–122: "the maximum …. peaked" sounds strange.*

We have exchanged "the maximum … peaked" with "the maximum … occurred"

*Line 125: as mentioned above, the release of piled-up waters normally occurs in the form of a (long) wave. This wave travels to the southwest while water velocities in it are fairly small (I guess on the order of 10 cm/s); thus "flow" is conceptually incorrect.*

We agree with the reviewer that this is badly phrased. The sentence now reads:

*"Consequently, the winds shifted from southwest to northeast, and the piled-up waters in the eastern Baltic Sea were released as a long wave travelling to the southwest."*

*Line 141: DWD was already explained.*

Thanks – revised.

*Line 152: check "methods … is described".*

Thanks – revised.

*Page 6, Caption to Fig. 1: check "forcing … are".*

Thanks – revised.

*Page 8, line 197: The water level was exceptionally high also in the Gulf of Finland. Soomere and Pindsoo (2016, Continental Shelf Research, 115, 53–64, doi: 10.1016/j.csr.2015.12.016) visualised modelled water levels above 80 cm near Tallinn for more than a week in March 1990.*

Thanks, we have added this to the text.

*Page 12, line 282: correct "Landort".*

Thanks – revised.

*Line 304: as above, it was motion of wave (energy), not really flow of water masses.*

We have rephrased:

*"The maximum water level at Landsort occurred as the piled-up waters were released and propagated south and westwards, reducing the water level in the north and east and causing it to rise throughout the southwestern Baltic."*

**Page 13, caption to Fig. 4: correct "capitol".**

Thanks – revised.

**Page 15, lines 338–343: I guess that this almost linear dependence may partially reflect the way how surface drag is calculated from the wind speed. I guess that readers would appreciate a short comment on that.**

We have added a comment on this and also refer back to another study where we discussed the parametrisation and formulation of the drag coefficient in the model setup more in detail. The section now reads:

*"Amplification of the wind speed resulted in increased peak water levels with an almost linear response (Fig. 6). This finding depends strongly on the model's wind stress parameterisation and drag coefficient, which was discussed in Andrée et al. (2022) for idealized simulations with the same model. The linear response seems to indicate that at least for the peak values, any dynamic changes to the sea-level patterns induced by the enhanced wind are marginal."*

**Page 17, lines 363–364: it may make sense to add that a decrease in salinity in the Baltic Sea may add to the sea level rise signal at the entrance of Danish straits.**

Thank you for this suggestion. There are multiple factors that contribute to the sea level rise signal, including salinity. We opted here not to highlight all components of sea level rise so as not to distract the audience. Rather, we include a reference to a local research on sea level rise (Su et al, 2021).

---

## Referee Report (RR1)

Dear Editor

I think the quality of the manuscript has much improved; the revised paper is much clearer.

However, please note there are a few comments that the Authors have not yet addressed:

- Line 6: the paragraph in the Authors' reply does not match that in the revised text, i.e., "historically observed circumstances" vs. "antecedent circumstances"
- From line 23 on: I still believe that the examples about wildfires and landslide are not relevant for the research, which is indeed about sea levels and coastal hazards. Anyway, if you want to leave those in the text, please at least add "[…] from days to months or even years, depending on the event considered"
- Figure 1 is too busy. I know I am being picky about this, but it is crucial that it is clearer to fully understand what is being discussed further on in the paper. Perhaps the close-up's size could be increased, and Authors could use different colors to label cities, islands, and straights
- Authors claim they modified Figure 1 (now Figure 2) according to my suggestions, but it doesn't look so. Same for Figure 5 (Figure 4 in the first version of the manuscript), which has not been modified
- Perhaps it would be easier to understand Figure 3 if Authors clearly specified why they used different preconditioning periods between the scenarios FLs and S in Sect. 2.2.1.

Besides, I still think that the English grammar still needs a thorough check before the paper is accepted for publication. See below a few typos:

- At line 4: you cannot say "may influence" as the event occurred in the past, rather you should say "[…] different prior conditions may had influenced peak […]"
- The term "precondition" is widely used and recurrent in the text, though I have never heard it before in the context of geophysical studies. As such, I wonder whether it is appropriate (if so, just overlook this comment and move ahead but please check this out)
- Line 8: was the increase modelled, or it was observed? You say that it "occurred" after saying "simulated", which is totally confusing
- Line 65: I am not sure that "propose" is well suited there. Perhaps "suggest"?
- Line 135: "during the original experiment, i.e., for" should be replaced with "for". Actually, you did not make an experiment, so the term is not appropriate
- Line 167: remove commas after "we" and "therefore"
- Line 309: this whole sentence is still unclear to me: "we assess the entire observation period (1886-2021) and each scenario for the occurrence pattern between elevated sea levels and their corresponding duration". What is that you are assessing?

I am not a native speaker so the list above is very limited, but as I said the whole text should be reviewed.

---

## Author Response (AR2)

Dear Reviewer,

We thank you for your willingness to review our manuscript and for your thorough and helpful comments on the manuscript, which once again has helped to improve on this work.

Please find our responses to your comments in blue below with the **line number in the revised manuscript**.

Best regards,

Elin Andrée, Jian Su, Morten Andreas Dahl Larsen, Martin Drews, Martin Stendel and Kristine Skovgaard Madsen

*I think the quality of the manuscript has much improved; the revised paper is much clearer.*

*However, please note there are a few comments that the Authors have not yet addressed:*

*- Line 6: the paragraph in the Authors' reply does not match that in the revised text, i.e., "historically observed circumstances" vs. "antecedent circumstances"*

We have revised the sentence, and are now using 'precondition circumstances'. Line 5-7.

*"We design numerical experiments by imposing a range of precondition circumstances as boundary conditions to numerical ocean model simulations. This allows us to quantify the change in peak water levels that arise due to alternative preconditioning of the sea level before the storm surge".*

*- From line 23 on: I still believe that the examples about wildfires and landslide are not relevant for the research, which is indeed about sea levels and coastal hazards. Anyway, if you want to leave those in the text, please at least add "[…] from days to months or even years, depending on the event considered"*

We have added the suggested sentence at Line 23.

*"The time scales of such "preconditioning" can vary from days to months or even years".*

*- Figure 1 is too busy. I know I am being picky about this, but it is crucial that it is clearer to fully understand what is being discussed further on in the paper. Perhaps the close-up's size could be increased, and Authors could use different colors to label cities, islands, and straights*

Thank you for this nice suggestion. As instructed, we have enlarged the size of the zoomed map and use different colours to label cities, islands and straits. Page 3.

*- Authors claim they modified Figure 1 (now Figure 2) according to my suggestions, but it doesn't look so. Same for Figure 5 (Figure 4 in the first version of the manuscript), which has not been modified*

Thank you for pointing them out again. We have now extended the colour bars and changed the marker colour of Landsort in Figures 2 and 5.

*- Perhaps it would be easier to understand Figure 3 if Authors clearly specified why they used different preconditioning periods between the scenarios FLs and S in Sect. 2.2.1.*

This is a very good point. In response, we now state the motivation of these three experiments at Line 202-204.

*"As previously mentioned, the filling level of the Baltic Sea in November 1872 was fairly moderate. To demonstrate the implications for extreme sea levels if the Baltic had been preconditioned differently, we formed scenarios by imposing the atmospheric forcing of 1872 onto three alternative cases where the sea-level patterns were different (Fig. 2)".*

*Besides, I still think that the English grammar still needs a thorough check before the paper is accepted for publication. See below a few typos:*

Thank you for pointing this out. All authors have helped thoroughly check the manuscript and improvements have been implemented as indicated in the track changes version (including the grammar items indicated below, which were specifically pointed out by the reviewer).

*- At line 4: you cannot say "may influence" as the event occurred in the past, rather you should say "[…] different prior conditions may had influenced peak […]"*

This has been changed. Line 4-5.

*"Here, we explore how various prior conditions could have influenced peak water levels for the devastating coastal flooding event in the western Baltic Sea in 1872".*

*- The term "precondition" is widely used and recurrent in the text, though I have never heard it before in the context of geophysical studies. As such, I wonder whether it is appropriate (if so, just overlook this comment and move ahead but please check this out)*

Thank you for this suggestion. We have checked and find that the word "precondition" is used as much as other phrases in the oceanographic and geophysical literatures. Therefore, we have kept the word "precondition" in the manuscript.

*- Line 8: was the increase modelled, or it was observed? You say that it "occurred" after saying "simulated", which is totally confusing*

We acknowledge that the sentence could be confusing, and have re-written it. Line 7-8.

*"As an example, a simulated increase of 36 cm compared to the 1872 event was seen in Køge".*

*- Line 65: I am not sure that "propose" is well suited there. Perhaps "suggest"?*

Reviewer's suggestion has been implemented. Line 65.

***- Line 135: "during the original experiment, i.e., for" should be replaced with "for". Actually, you did not make an experiment, so the term is not appropriate***

Thank you for pointing this out. Accordingly, the text has been changed, so we do not refer to the "original experiment" but to the "reference simulation". Line 135-136.

*"The following section describes the atmospheric conditions during the reference simulation, i.e., for the unperturbed simulation of the 1872 storm surge as reconstructed by our model system. We denote this simulation O".*

***- Line 167: remove commas after "we" and "therefore"***

Done.

***- Line 309: this whole sentence is still unclear to me: "we assess the entire observation period (1886-2021) and each scenario for the occurrence pattern between elevated sea levels and their corresponding duration". What is that you are assessing?***

This has been clarified. Line 311-312.

*"We analyzed the occurrence pattern between elevated sea levels and their corresponding duration for the entire observation period (1886-2021").*